# Extracts from Fermented Black Garlic Exhibit a Hepatoprotective Effect on Acute Hepatic Injury

**DOI:** 10.3390/molecules24061112

**Published:** 2019-03-20

**Authors:** Jen-Chieh Tsai, Yi-An Chen, Jung-Tsung Wu, Kuan-Chen Cheng, Ping-Shan Lai, Keng-Fan Liu, Yung-Kai Lin, Yu-Tsang Huang, Chang-Wei Hsieh

**Affiliations:** 1Department of Medicinal Botanicals and Health Applications, Da-Yeh University, Chang-Hua 51591, Taiwan; jenchieh@mail.dyu.edu.tw; 2College of Biotechnology and Bioresources, Da-Yeh University, Chang-Hua 51591, Taiwan; anncerita@gmail.com (Y.-A.C.); chen.ben495@gmail.com (J.-T.W.); 3Institute of Biotechnology, National Taiwan University, Taipei 10617, Taiwan; kccheng@ntu.edu.tw; 4Graduate Institute of Food Science Technology, National Taiwan University, Taipei 10617, Taiwan; 5Department of Chemistry, National Chung Hsing University, Taichung 402, Taiwan; pslai@email.nchu.edu.tw; 6School of Chinese Pharmaceutical Sciences and Chinese Medicine Resources, College of Pharmacy, China Medical University, Taichung 40402, Taiwan; stella89132@yahoo.com.tw; 7Global Research and Industry Alliance Center, National Chung Hsing University, Taichung 402, Taiwan; nitrite.tw@gmail.com; 8Graduate Institute of Biomedical Engineering, National Chung Hsing University, Taichung 402, Taiwan; 9Menglinbeier Clinic, Taichung 407, Taiwan; yuhchanghuang@gmail.com; 10Department of Food science and biotechnology, National Chung Hsing University, Taichung 402, Taiwan; 11Department of Medical Research, China Medical University Hospital, Taichung 404, Taiwan

**Keywords:** black garlic, acute hepatic injury, carbon tetrachloride, antioxidant, anti-inflammatory

## Abstract

The mechanism of hepatoprotective compounds is usually related to its antioxidant or anti-inflammatory effects. Black garlic is produced from garlic by heat treatment and its anti-inflammatory activity has been previously reported. Therefore, the aim of this study was to investigate the hepatoprotective effect of five different extracts of black garlic against carbon tetrachloride (CCl_4_)-induced acute hepatic injury (AHI). In this study, mice in the control, CCl_4_, silymarin, and black garlic groups were orally administered distilled water, silymarin, and different fraction extracts of black garlic, respectively, after CCl_4_ was injected intraperitoneally to induce AHI. The results revealed that the *n*-butanol layer extract (BA) and water layer extract (WS) demonstrated a hepatoprotective effect by reducing the levels of alanine aminotransferase (AST), alanine transaminase (ALT), alkaline phosphatase (ALP), and hepatic malondialdehyde (MDA). Furthermore, the BA and WS fractions of black garlic extract increased the activity of superoxide dismutase (SOD), glutathione peroxidase (GSH-Px), glutathione reductase (GSH-Rd), tumor necrosis factor alpha (TNF-α), and the interleukin-1 (IL-1β) level in liver. It was concluded that black garlic exhibited significant protective effects on CCl_4_-induced acute hepatic injury.

## 1. Introduction

Garlic (*Allium sativum* L.) has been used as a spice and traditional medicine for centuries. Many studies have shown that garlic has numerous beneficial effects for human health including antioxidant, anti-inflammation, anti-cancer, lipid regulation, reduction of blood pressure, and improvement of blood glucose control [1,2,3,4,5,6]. Additionally, garlic and its derivatives exhibit hepatoprotective effects against alcoholic hepatic injury, drug-induced hepatic injury, fibrogenic liver disorders, etc. [7]. For instance, to improve hypoglycemic and hypolipidemic activities, taking 350 mg/kg body weight of garlic extract can significantly decrease the levels of blood glucose, triglycerides, and cholesterol in diabetic rabbits [8]; for the hepatoprotective effect, male albino rats treated with heated garlic juice at 100 mg/kg/day for four weeks could suppress hepatic oxidative stress via the Nrf2/Keap1 pathway [9].

However, the consumption of unprocessed raw garlic is limited due to its characteristic odor and spicy taste, causing the gastric mucosal cells injury with excessive intake [10]. Fermented black garlic, a kind of heat processed garlic, has been reported for its biological activities such as anti-cancer activity, anti-obesity activity, hepatoprotective activity, and anti-inflammatory activity [1,6]. For anti-cancer activity, 100 mg/mL age black garlic (ABG) extract could induce apoptosis in human gastric cancer cells (SGC-7901) [11]; for anti-obesity activity, the mice given 400 mg/kg BG significantly decreased in body weight, abdominal fat weight, abdominal adipocyte diameters, and abdominal fat pad thickness [12]; and 200 mg/kg ABG demonstrated hepatoprotective activity by decreasing ALT and AST levels in the liver of Sprague–Dawley rats [13]. Nevertheless, knowledge of the ingredients or compounds possessing those bioactivities remains unclear.

In our previous study, five solvents (*n*-hexane, dichloromethane, ethyl acetate, *n*-butanol, and water) were used to separate compounds with different polarity in black garlic [1]. Results showed that black garlic promoted gastrointestinal motility mainly in the *n*-butanol and water fractions. In this study, a similar system was adopted to evaluate whether black garlic extracts may have a hepatoprotective effect.

## 2. Results

### 2.1. Measurement of Serum AST and Serum ALT

According to the previous study, we found extremely a low extraction efficiency of n-hexane and dichloromethane (extraction rates: water > *n*-butanol > ethyl acetate > dichloromethane > n-hexane) [1]. However, which components of black garlic exhibit a hepatoprotective effect have never been investigated. Thus, our study was designed to use the three fractions of ethyl acetate layer extract (EA), *n*-butanol layer extract (BA), and water layer extract (WS) to assess the conditions of hepatoprotection in vivo serum AST and serum ALT level, as shown in Figure 1, by using silymarin as a positive control group, which is now widely used to treat hepatic injury. The carbon tetrachloride (CCl_4_) group exhibited a significant increase in AST, ALT, and alkaline phosphatase (ALP), leading to severe hepatic injury. However, the increase of ALT and AST levels were significantly inhibited by treatment with BA and WS. There were no significant differences in the EA group. These results suggest that the different extraction solutions of black garlic provided protective properties against CCl_4_-induced hepatic injury.

### 2.2. Histopathological Analysis 

To evaluate the major pathological changes based on the increased vacuolization, inflammation, and coagulated necrosis, four grades were categorized in order to present the level of hepatic injury. The hepatic histological results are shown in Figure 2. Hepatic injuries were reduced by treatment with BA and WS (0.2 and 0.5 g/kg) (Figure 2D–G) and the histological index of vacuolization and hepatocellular necrosis of the liver were significantly decreased. Our histological outcome also demonstrated that treatment with BA and WS significantly ameliorated CCl_4_-induced hepatic injury.

### 2.3. Hepatic TNF-α and IL-1β Assays

TNF-α and IL-1β in CCl_4_-induced acute hepatic injury. The effects of black garlic extraction (BA and WS) on CCl_4_-induced lipid peroxidation are shown in Figure 3. The level of TNF-α and IL-1β in the CCl_4_ group was significantly increased (*p* < 0.001); however, TNF-α and IL-1β were significantly reduced by treatment with BA and WS at 0.5 g/kg (*p* < 0.001).

### 2.4. Hepatic MDA Assay

Quantification of MDA is important for assessing hepatic lipid peroxidation. The effects of black garlic extraction (BA and WS) on CCl_4_-induced lipid peroxidation are shown in Figure 4. The level of MDA in the CCl_4_ group was significantly increased (*p* < 0.001); however, the MDA levels were significantly reduced by treatment with BA at 0.2 g/kg (*p* < 0.01) and 0.5 g/kg (*p* < 0.001), silymarin (200 mg/kg) (*p* < 0.001), and WS (0.2 and 0.5 g/kg) (*p* < 0.01).

### 2.5. Measurement of Hepatic Glutathione (GSH) Level

GSH is an important non-enzymatic antioxidant compound. Oxidizing GSH to glutathione disulfide (GSSG) will reduce hydrogen peroxide, hydroperoxide, and xenobiotic toxicity. The effects of black garlic extract on GSH levels are shown in Figure 5. CCl_4_ treatment led to a significant reduction of GSH level when compared to the normal group (*p* < 0.001). However, the GSH levels were increased significantly by treatment with WS at 0.2 g/kg (*p* < 0.05) and 0.5 g/kg (*p* < 0.001), BA (0.2 and 0.5 g/kg) (*p* < 0.05), along with silymarin at 200 mg/kg (*p* < 0.01).

### 2.6. Measurement of Antioxidant Enzymatic Activity

CAT, SOD, GSH-Px, and GSH-Rd were measured to access the antioxidant effects of black garlic extract. The hepatic enzyme activities of the CCl_4_ group was significantly lower than that of the control group (Figure 6). Treatment with BA and WS at doses of 0.2 g/kg and 0.5 g/kg and silymarin at a dose of 200 mg/kg increased the levels of SOD, GSH-Px, GSH-Rd, and glutathione S-transferase (GST) activity. However, BA at doses of 0.2 g/kg did not significantly increase the levels of CAT.

## 3. Discussion

The development of fibrosis, particularly cirrhosis, is associated with morbidity and mortality [14]. An animal model of hepatic injury induced by CCl_4_ has been commonly used to study the hepatoprotective effects of natural medicine, and refers to an ordinary method for the investigation of hepatoprotective activity of different kinds of medicine [15,16,17].

The increase in serum AST and ALT levels was attributed to impaired liver structure. Generally, cell deaths were associated with plasma permeability as the levels of AST and ALT became outstanding due to hepatic structure damage [16,18]. As a result, hepatic injury can be determined by analyzing serum AST and ALT levels. In our study, the levels of AST, ALT, and ALP in serum increased after CCl_4_ was introduced. This outcome can be attributed to hepatic damage, resulting in an increased rate of synthesis or release of functional enzymes from biomembranes [19]. Usually, ALT and AST are used to assess liver function; ALT is very accurate in monitoring hepatocellular status, and AST is a sensitive indicator of mitochondrial problems, especially in centrilobular areas of the liver [20]. Lipid peroxidation has been categorized as one of the most important causes of CCl_4_-induced hepatic injury [21]. 

Inflammation was triggered by CCl_4_-induced hepatotoxicity, followed by the release of proinflammatory cytokine such as IL-lβ and TNF-α. IL-1β and TNF-α are key cytokines in inflammation and the content of IL-1β and TNF-α increases during the development of hepatic injury [22]. IL-1β is a proinflammatory cytokine that plays an important role in regulating liver NO synthesis [23]. TNF-α is produced by Kupffer cells, activates T cells and macrophages, and stimulates secretion of other inflammatory cytokines and NO. Our results indicated that the production of TNF-α and IL-1β was significantly reduced after feeding the BA and WS layers, demonstrating that BA and WS have a hepatoprotective effect on CCl_4_-induced damage. MDA was formed due to free radicals attacking the plasma membrane and refers to the end product of lipid peroxidation. In addition, MDA has been widely used as an indicator of lipid peroxidation damage [24,25]. 

In the latest study, an increase in the hepatic MDA level suggested the enhancement of lipid peroxidation, consequently leading to hepatic damage as well as the inactivation of the antioxidant defense system. To estimate the effect of BA and WS pretreatment on CCl_4_-induced liver lipid peroxidation, MDA was monitored. In the study, MDA levels were significantly reduced by treatment with BA at 0.2 g/kg and 0.5 g/kg, silymarin (200 mg/kg), and WS (0.2 and 0.5 g/kg). 

Reactive oxygen species (ROS) formed in the CCl_4_-induced hepatic damage model could significantly decrease through the expressions of antioxidant enzymes such as SOD, GSH-Px, and GSH-Rd. SOD, GSH-Px, and GSH-Rd are antioxidative enzymes that are known to be easily inactivated by lipid peroxides or ROS when CCl_4_ is administrated [16]. Therefore, an increase in antioxidant activity and the inhibition of free radical generation is positively correlated with hepatic protection. GSH is a main non-enzymatic anti-oxidant that decreases hydrogen peroxide, hydroperoxide, and xenobiotic toxicity through the oxidation of GSH transforming to GSSG [26]. GSH content was significantly decreased, which is an important indicator of CCl_4_ toxicity due to CCl_4_ administration being prone to reduce the levels of GSH. In this study, the data showed that the level of GSH was increased significantly by treatment with BA, WS, and silymarin. 

In the enzymatic defense system, SOD dissimulates superoxide to H_2_O_2_ in order to prevent oxygen toxicity. Both GSH-Px and GSH-Rd are GSH-related enzymes and play a role in detoxification, while antioxidant activity in cellular defense undergoes conjugation with glutathione or the reduction of free radicals [24,27,28]. GSH-Px works with GSH to metabolize H_2_O_2_, which allows a harmful toxin to become a nontoxic product, while GSH-Rd catalyzes the reduction of GSSG to GSH. Our results revealed that the activities of SOD, GSH-Px, and GSH-Rd were significantly decreased during the development of CCl_4_-induced acute hepatic injury [29], the outcome of which agreed with those of previous studies. In this study, the activities of SOD, GSH-Px, and GSH-Rd improved after treatment with BA and WS. These results suggest that BA and WS reduce ROS production by increasing hepatic antioxidant activities and against hepatotoxicity in order to provide hepatic protection [25]. The suppression of MDA production is likely to promote the activities of SOD, GSH-Px, and GSH-Rd. Furthermore, an increase in SOD activity not only increases the superoxide anion scavenging capacity, but also prevents peroxynitrite production. 

The present study suggests that the black garlic extract with *n*-butanol fraction can exhibit hepatoprotective activity to attenuate CCl_4_-induced acute hepatic injury in mice, and this result was proven by the decreased AST and ALT levels and liver histopathological analysis. There was a lack of significant difference in EA treatment. The hepatoprotective mechanisms of black garlic extraction were likely to be related to the decreased MDA level and increased GSH level by increasing the activities of anti-oxidative enzymes such as SOD, GSH-Px, GSH-Rd, and GST. In addition, BA increased the physiological antioxidant activities to render a reduction of hepatic injury caused by free radical (ROS and peroxide) attacks during the metabolism of CCl_4_. In our research results, both BA and WS could significantly decrease CCl_4_-induced acute hepatic injury, which caused the reduction in the content of ALT, AST, and MDA, and also significantly decreased liver inflammatory response. Moreover, in our research, black garlic exhibited a hepatoprotective effect by increasing antioxidant enzymatic activity (SOD, GSH-Px, and GSH-Rd) and increasing the content of CAT as well as decreasing CCl_4_-induced acute hepatic injury.

Furthermore, the flavor, nutritional, and functional properties of the garlic change during the thermal processing of black garlic by the Maillard reaction [30] as black garlic produced in a high temperature, humid environment matures and reduces the spicy and irritation qualities from raw garlic under control conditions [31]. γ-glutamyl-S-allyl-L-cysteines (GSACs) and S-allyl-L-cysteine sulfoxide (alliin) are the major sulfur-containing compounds in intact garlic [32]. After cutting or squeezing, alliinase can be released from the garlic, which hydrolyzes alliin to allicin. However, allicin is an unstable compound; it decomposes to yield organic lipid-sulfur-containing compounds, e.g., diallyl sulfide (DAS), diallyl disulfide (DADS), diallyl trisulfide (DATS), and allyl methyl sulfide (AMS), ajoene. Moreover, water-soluble sulfur compounds (S-allylcysteine, SAC) are also formed [5,33,34,35]. Previous studies have shown that the SAC content in black garlic is six-fold higher than that in fresh garlic. Studies have demonstrated that SAC improved liver disease via regulation of the peroxisomal proliferator activator receptor α (PPAR-α), sterol regulatory element binding protein 1c (SREBP-1c), serum adiponectin levels, liver MDA, reactive oxygen species (ROS) content, serum ALT, serum AST, GSH content, GSH-Px and CAT activities, and the inhibition of lipolysis. Additionally, SAC can significantly inhibit N-nitrosodiethylamine (NDEA)-induced hepatocarcinogenesis [4,36,37,38]. We further analyzed the SAC content of EA, BA, and WS where, from the data, it was found that SAC only appeared in BA (497.58 ± 43.78 μg/g, Table 1).

Therefore, we surmise that SAC is a major active compound in BA [1,39]. However, there was no SAC in WS, but our result showed that WS had a hepatoprotective effect. In previous research, we found that WS has abundant polysaccharides as well as great antioxidant activities, e.g., DPPH radical scavenging ability. Thus, we surmised that the main ingredients of hepatoprotective effect in black garlic might be SAC and polysaccharides that inhibit CCl_4_-induced hepatic injury by inhibiting lipid peroxidation and inflammation.

## 4. Materials and Methods

### 4.1. Materials

ICR mice (20–25 g; 8 weeks) were purchased from Bio LASCO Taiwan Co. Ltd. (Taipei, Taiwan). These were placed in a standard cage with a relative humidity of 55 ± 5%, a 12 h/12 h light dark cycle, and a constant temperature of 21 ± 2 °C for at least one week and fed food and water ad libitum. All animal procedures were conducted in accordance with the standards set forth in the guidelines for the Care and Use of Experimental Animals by the Committee for the Purpose of Control and Supervision of Experiments on Animals and the National Institutes of Health. The protocol was approved by the Committee on Animal Research, Da-Yeh University, under code 104007. The black garlic was provided by All Wealth Biotech Co. Ltd. (Xiamen, China). All of the other chemicals used in these studies were of analytical grade and obtained commercially.

### 4.2. Preparation of Black Garlic

The black garlic was chopped, weighed, and extracted by solvents with different polarities in the following sequence: *n*-hexane, dichloromethane, ethyl acetate, *n*-butanol, and distilled water [1]. First, black garlic was extracted by five times volume of *n*-hexane, and filtrated with filter paper (90 mm diameter) after the extraction of *n*-hexane; this step was repeated three times. When the first-round extraction was completed, the solvent was changed to dichloromethane for continued extraction, with the steps of extraction as previously described; the process was continued according to this regular procedure and five different extract solutions were obtained. The partition extraction rate (E) was calculated as follows (Equation (1)):(1)$$E=\frac{V_{t}}{C_{t}}\times 100\%$$
where V_t_ is the different polar solvents extract weight, and C_t_ is the sample (before extraction) weight.

### 4.3. Carbon Tetrachloride Induced Acute Hepatic Injury

Experimental animals were randomly divided into 8 experimental groups (10 mice per group). Mice in the control group and CCl_4_ group were fed with distilled water. Silymarin was orally administered in the mice in the silymarin group (200 mg/kg in 1% carboxymethylcellulose), while mice in the EA groups, BA groups, and WS groups were orally administered a fractional extract of black garlic (0.2 g/kg and 0.5 g/kg) for seven consecutive days. One hour after the last administration of the experimental reagent, CCl_4_ (10 mL/kg, 0.2% in olive oil) was injected intraperitoneally into each group of mice, except for the normal control group. The control mice received an equivalent volume of olive oil (i.p.) [40].

### 4.4. Preparation of Blood and Liver

Twenty-four hours after the CCl_4_-injection, mice were sacrificed under anesthesia. Blood was collected for the evaluation of the biochemical parameters (AST, ALT). The right lobe of the liver was resected and fixed with 4% paraformaldehyde and embedded in paraffin for histological analysis; other hepatic tissue were diluted by saline (1:1), homogenated, and centrifuged. The supernatant was then taken as the homogenate hepatic sample, and stored at −80 °C in order to analyze the activity of a series hepatic enzyme [41].

### 4.5. Serum Biochemistry Analysis for Hepatic Function

Blood samples were collected from the mice and coagulated under room temperature for 1 h. The plasma was prepared by centrifugation at 3000 rpm, 4 °C for 10 min. An auto-analyzer (Hitachi 7060, Hitachi, Tokyo, Japan) was used to analyze and determine the AST and ALT [42].

### 4.6. Pathological Analysis

All mice were sacrificed. One piece of liver tissue (1 × 1 × 1 cm^3^) was cut from the largest right lobe, and fixed with formalin and processed with hematoxylin-eosin (H & E) staining for histological preparation to evaluate the hepatic damage [42].

### 4.7. Hepatic TNF-α and IL-1β Assays

TNF-α and IL-1β were analyzed using ELISA kits according to the manufacturer’s instructions. The captured antibodies of TNF-α and IL-1β were added to a 96-well plate overnight. The next day, a biotinylated antibody was added before incubation with sample tissues or standard antigens. Finally, biotinylated antibodies were incubated with sample tissues or standard antigens in the plate before streptavidin-horseradish peroxidase (HRP) was added to end the reaction. The amounts of TNF-α and IL-1β were represented as pg/mg protein at 450 nm [18].

### 4.8. Hepatic Malondialdehyde (MDA) Assay

Thiobarbituric acid reactive substance (TBARS) content was analyzed to assess the MDA level. [39]. MDA is a naturally occurring product of lipid peroxidation. Briefly, MDA reacted with thiobarbituric acid at high temperature under an acidic condition and presented a red-complex TBARS. The TBARS was determined colorimetrically at 532 nm.

### 4.9. Measurement of Hepatic GSH Level

Liver homogenate (50 µL) of each sample was mixed with 10% trichloroacetic acid (TCA) (50 μL) and then placed on ice. After centrifugation (10,000 g for 5 min), the supernatant (330 µL) was taken and normal saline (600 µL), along with 5,5′-dithio-bis-(2-nitrobenzoic acid) (DTNB) (125 µL) were added and mixed well. The absorbance was read at 412 nm. The standard curve was established using glutathione. The concentration of GSH was expressed as μmol/g protein in the sample [18,42].

### 4.10. Antioxidant Enzymatic Activity Measurements

Liver homogenates were prepared by using a homogenizer to determine CAT, SOD, GSH-Rd, and GSH-Px activities. These antioxidant enzymatic activities were determined using the kits (Randox Laboratories Ltd., Crumlin, UK) according to the manufacturer’s instructions and detected with the Chem Well^®^-T Automated Chemistry Analyzer (Wicklow, Ireland). The CAT results were expressed in units per milligram of protein (nmol/mg protein), along with the results of SOD, GSH-Px, and GSH-Rd expressed in units per milligram of protein (U/mg protein) [43].

### 4.11. Quantification of SAC in Black Garlic

The black garlic samples (5 ± 0.5 g) were added to 50 mL distilled water. The mixture was extracted by ultrasonic processing for 10 min. The processed solution was filtered through 0.45 μm and 0.22 μm syringe filters. The filtered samples were subjected to HPLC with an ultraviolet detector (L-7400, Hitachi, City, Japan). Separation of the sample was conducted with a C18 column (250 mm × 4.6 mm ID, 5 μm; Waters). The flow rate was 0.6 mL/min, the sample injected velum was 10 μL, and the detecting wavelength was 210 nm. The solvent system consisted of distilled acetonitrile and water (12:88, *v*/*v*) [44,45].

### 4.12. Statistical Analysis

Values were expressed as mean ± standard deviation (SD). To evaluate the differences between the groups studied, one-way analysis of variance (ANOVA) with Scheffe’s multiple range post hoc tests was used. A non-parametric (Kruskal–Wallis’s test) followed by a Mann–Whitney U-test was used to determine the statistical differences among the groups for histopathological analysis. Differences were considered statistically significant at a *p*-level of 0.05.

## Figures and Tables

**Figure 1 molecules-24-01112-f001:**
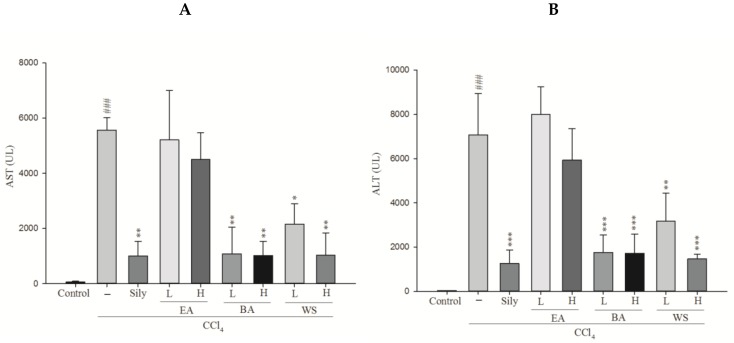
Effect of different fractions of black garlic and silymarin on serum (**A**) AST (**B**) ALT activities in mice treated with CCl_4_. Each value was represented as mean ± S.E.M. (*n* = 10). ^###^
*p* < 0.001 as compared with the control group. * *p* < 0.05, ** *p* < 0.01, *** *p* < 0.001 as compared with the CCl_4_ group (one-way ANOVA followed by Scheffe’s multiple range test).

**Figure 2 molecules-24-01112-f002:**
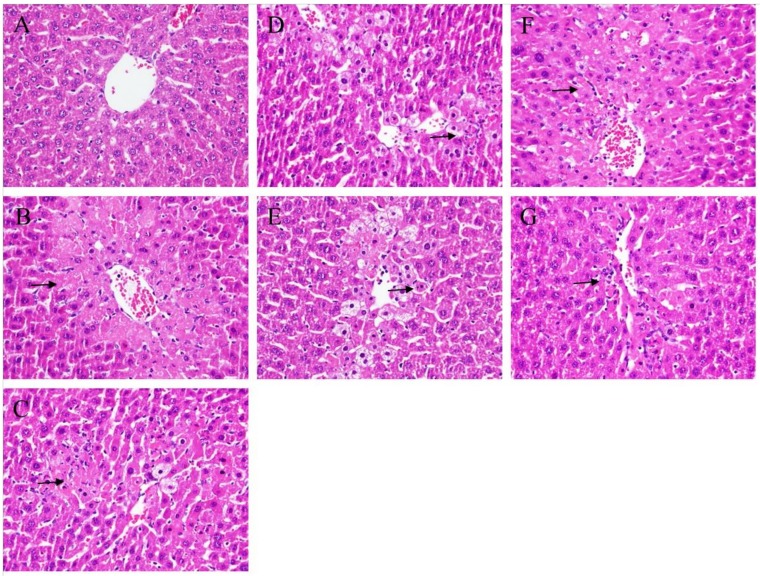
Hepatichistological analyses to the effect of different fractions by black garlic and silymarin on CCl_4_-induced acute liver damage in mice. Liver tissues were stained with H & E (400X). (**A**) control group; (**B**) animals treated with CCl_4_ displayed cell necrosis (short arrow); (**C**) animals pre-treated with silymarin and then treated with CCl_4_; (**D**) animals treated with BA (0.2 g/kg), and then treated with CCl_4._; (**E**) animals treated with BA (0.5 g/kg), and then treated with CCl_4_; (**F**) animals treated with WS (0.2 g/kg), and then treated with CCl_4_; (**G**) animals treated with WS (0.5 g/kg), and then treated with CCl_4_. BA: *n*-butanol layer extract; WS: water layer extract

**Figure 3 molecules-24-01112-f003:**
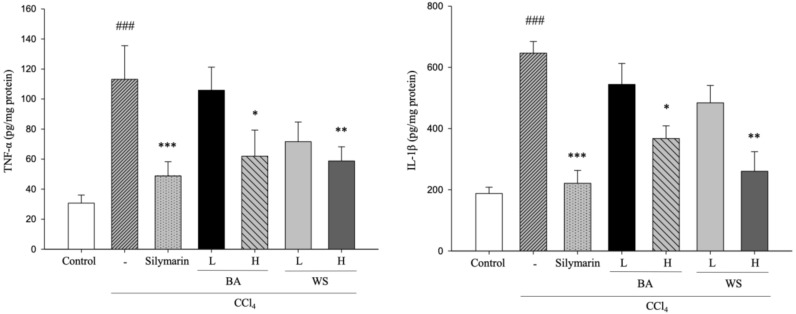
Effect of different fractions of black garlic and silymarin on hepatic TNF-α and IL-1β in mice treated with CCl_4_. Each value represents as mean ± S.E.M. (*n* = 10). ^###^
*p* < 0.001 as compared with the control group. * *p* < 0.05, ** *p* < 0.01, *** *p* < 0.001 as compared with the CCl_4_ group (one-way ANOVA followed by Scheffe’s multiple range test).

**Figure 4 molecules-24-01112-f004:**
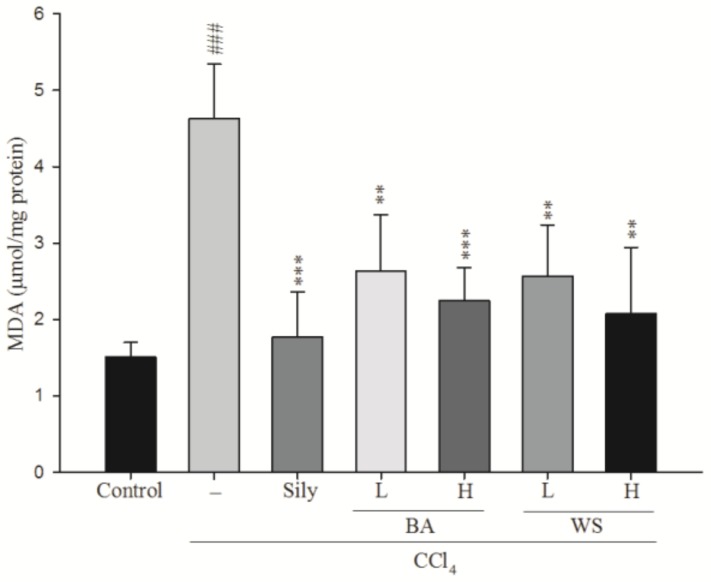
Effect of different fractions of black garlic and silymarin on hepatic MDA content in mice treated with CCl_4_. Each value represents as mean ± S.E.M. (*n* = 10). ^##^
*p* < 0.01, ^###^
*p* < 0.001 as compared with the control group. ** *p* < 0.01, *** *p* < 0.001 as compared with the CCl_4_ group (one-way ANOVA followed by Scheffe’s multiple range test).

**Figure 5 molecules-24-01112-f005:**
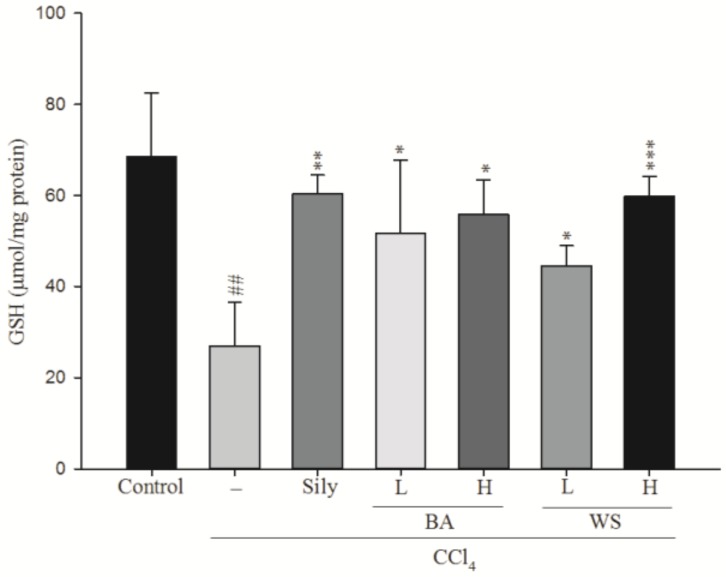
Effect of different fractions of black garlic and silymarin on hepatic GSH content in mice treated with CCl_4_. Each value represents as mean ± S.E.M. (*n* = 10). ^##^
*p* < 0.01 as compared with the control group. * *p* < 0.05, ** *p* < 0.01, *** *p* < 0.001 as compared with the CCl_4_ group (one-way ANOVA followed by Scheffe’s multiple range test).

**Figure 6 molecules-24-01112-f006:**
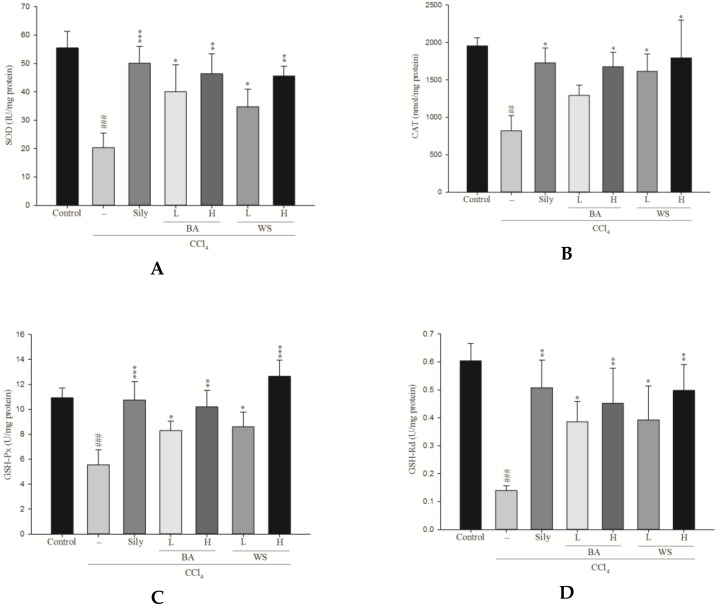
Effect of different fractions of black garlic and silymarin on hepatic (**A**) SOD (**B**) CAT (**C**) GSH-Px (**D**) GSH-Rd activities in mice treated with CCl_4_. Each value represents as mean ± S.E.M. (*n* = 10). ^##^
*p* < 0.01, ^###^
*p* < 0.001 as compared with the control group. * *p* < 0.05, ** *p* < 0.01, *** *p* < 0.001 as compared with the CCl_4_ group (one-way ANOVA followed by Scheffe’s multiple range test).

**Table 1 molecules-24-01112-t001:** SAC content (μg/g dry weight) of EA, BA and WS fractions.

Extract Solvent	SAC Content (μg/g)
EA	ND
BA	497.58 ± 43.78
WS	ND

ND means not detected.

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
