# Peer review of "Extracts from Fermented Black Garlic Exhibit a Hepatoprotective Effect on Acute Hepatic Injury"

_molecules, 2019, doi:10.3390/molecules24061112_

Round 1
Reviewer 1 Report
The manuscript investigated the protective effects and mechanisms of different extractions of Black garlic against CCL4-induced liver injury. The studies are well-designed in general and addressed the important unanswered questions in the field and has potential in clinical application. The data are all in good quality and supports the conclusion. Detailed critics and suggestions are listed as below.
The author did not mention the significance of the “silymarin” group throughout the manuscript. Is this an agent with known hepatoprotective activity? If so, it should be more clearly stated.
In legend of Figure2, it is unclear which of panel D-G are liver from mice treated with BA and which are treated with WS.
Titles of section 2.5 and 2.6 in the results have the same title. Please double check. Also, the titles of the results section segments have a heavy focus on the method used. Those should emphasize more on describing the observations from the experiments.
The data strongly support that BA and WS fraction of the black garlic is protective against liver injury and inflammation possibly through antagonizing ROS production. It is not required by would be very interesting to see if BA and WS has additive effects in their hepatoprotection, which would indicate distinctive mechanism for their function.
The language of the manuscript is in good quality. But there are occasional sentences difficult to follow. The reviewer recommend a thorough proof read.
Author Response
We appreciate the reviewer's comments very much, and have revised the manuscript with red font color. The specific changes we have made are in the attachment.

Reviewer 2 Report
This manuscript by Tsai et al., shows beneficial function of black garlic extract against CCl4-induced liver injury and inflammation.
Major comments:
1. How one dose of CCL4 increased ALT or AST to 6000 u/L, is not clear to me. It is too high. The histopathology shown by hematoxylin-Eosin staining did not show that much liver injury. Please check the calculations or the kit, something must be wrong.
2. It was not clear how authors standardized the black garlic preparation. Method of this preparation could be variable. Authors should measure some specific standard component in these extracts, such as polyphenol content, to standardize the dose in different experiments.
3. The histopathological analysis is not clear. Images did not justify the results.
4. GSH assay: DTNB assay measures total thiol not GSH. Authors need to use a more specific method such as Glutathione reductase-coupled assay for measuring reduced GSH.
Author Response

(The authors gave the same response as above.)

Reviewer 3 Report
Authors used animal models to study the hepatoprotective effect of the fermented black garlic extracts. The results indicated that black garlic had significant protective effect on CCl4 induced acute hepatic injury. The experiment was well designed and results were discussed in details. Authors should read the manuscript carefully and correct all the spelling and grammar errors. Past tense should be used when experiments were discussed.
A few comments are listed a bellows.
1. Abbreviations were used in the whole manuscript. Please take not that when abbreviation was appeared first time in the manuscript, the full name should be given. It will be good if all the abbreviations can be complied in the list for easy reference.
2. Line 2, exhibit
3. Line 62, body weight
4. Line 73-75, delete the three lines
5. Line 80, incomplete sentence, please revise
6. Line 131. GHS is an important non-enzymatic antioxidant compound. Oxidizing GSH to glutathione bisulfide (GSSG) will reduce hydrogen peroxide, hydroperoxide and xenobiotic toxicity.
7. Line 143, please change the title of this paragraph.
8. Line 184 and 185. The meaning of the sentences is very confusing. Please revise.
9. Line 196. …GSH raised…
10. Line 201, …while GRd catalyzes…
11. Line 217, Don’t understand what the sentence means. Please revise “Mechanism of black garlic extract…”
12. Line 231-232, …was degradable or volatile at high temperature.
13. Line 234-235, please revise the sentence to make it clear.
14. Line 236, delete mainly
15. Line 245, …polysaccharides which inhibited…
16. Line 246, …through inhibiting…
17. Line 253, …ad libitum…
18. Line 261, …was chopped, weighed and extracted by solvents with different polarities in the following sequence.
19. Line 263, please explain what is “thermally filtrated”.
20. Line 267, …different extract solutions
21. Line 270, please explain what is “weight in phase” and “sample weight phase volume”.
22. Line 299, …, fixed with formalin and processed…
23. Line 308, delete “to determine their amount”
24. Line 311, …was analyzed…
25. Line 313, …at high temperature under…
26. Line 319, delete “ as added and mixed well”
27. Line 320, replace “which was” with “were”.
28. Line 321. …standard curve was established using”
29. Line 328, …instructions and detected with…
30. Line 329, … in µmol /g protein…; in U / mg protein.
31. Line 346, … at p level of 0.05.
Author Response

(The authors gave the same response as above.)
